# Progression of carotid intima-media thickness, visceral fat accumulation and metabolic derangement in people living with HIV initiating antiretroviral therapy: A prospective cohort study at Thailand's tertiary care center

Thirajit Boonsaen[1,*], Winai Ratanasuwan[1‡], Boonrat Tassaneetrithep[2‡], Tanyaluck Thientunyakit[3‡], Jitsupa Wongsripuemtet[3‡], Shanigarn Thiravit[3‡], Phatharajit Phatharodom[1‡], Oranich Navanukroh[1‡], Mayuree Homsanit[1]

1 Department of Preventive and Social Medicine, Faculty of Medicine Siriraj Hospital, Mahidol University, Bangkok, Thailand, 2 Center of Research Excellence in Immunoregulation, Faculty of Medicine Siriraj Hospital, Mahidol University, Bangkok, Thailand, 3 Department of Radiology, Faculty of Medicine Siriraj Hospital, Mahidol University, Bangkok, Thailand

☯ These authors contributed equally to this work.
‡ WR, BT, TT, JW, ST, PP and ON also contributed equally to this work.
* thirajit.boo@mahidol.ac.th

## Abstract

### Background

Combination antiretroviral therapy (ART) has extended life expectancy for people with HIV, but long-term treatment is associated with adverse changes in body composition and cardiovascular risk. We evaluated 36-month changes in adiposity, metabolic parameters, and carotid intima-media thickness (cIMT) in Thai adults initiating ART.

### Methods

A prospective cohort of 132 ART-naïve adults was followed for 36 months. Assessments at baseline, 12, 24, and 36 months included anthropometry; body composition by bioelectrical impedance analysis (BIA) and dual-energy X-ray absorptiometry (DEXA); metabolic and renal indices; and bilateral carotid ultrasound. Longitudinal changes were analyzed using non-parametric tests, and correlations were examined between cIMT and body composition or metabolic measures.

### Results

Virological suppression exceeded 90% and CD4 counts improved steadily. However, notable adiposity changes were observed. Median BMI and waist circumference increased (both $p < 0.01$); total fat mass rose by 6.7%; visceral adipose tissue (VAT) increased by 33%; and the android/gynoid ratio exceeded 1.0 by 24

**Data availability statement:** All relevant data are contained within the paper and its Supporting information files. Individual-level data used in this study are not publicly available due to ethical restrictions involving patient confidentiality. However, de-identified data may be made available upon reasonable request to qualified researchers who meet the criteria for access to confidential data, subject to approval by the Siriraj Institutional Review Board (SIRB), Faculty of Medicine Siriraj Hospital, Mahidol University. Contact Information: Siriraj Institutional Review Board (SIRB) Faculty of Medicine Siriraj Hospital, Mahidol University 2 Wang Lang Road, Bangkoknoi, Bangkok 10700, Thailand Tel: +66 2 419 2617 Email: sirb@mahidol.ac.th.

**Funding:** This work was supported by the Research Development Grant, Faculty of Medicine Siriraj Hospital, Mahidol University (Research Project Code (IO): R016034002). The funders had no role in study design, data collection and analysis, decision to publish, or preparation of the manuscript.

**Competing interests:** The authors have declared that no competing interests exist.

months, reflecting central fat redistribution. Fasting glucose increased ($p < 0.05$) while HOMA-β declined ($p < 0.05$), indicating early β-cell dysfunction; lipid profiles remained stable. Mean cIMT increased across arterial segments, most prominently at the carotid bifurcations (right: 0.644 mm at baseline to 0.729 mm at 36 months; + 0.085 mm; left: 0.675 mm at baseline to 0.756 mm at 36 months; + 0.081 mm both $p < 0.01$). Right bifurcation cIMT correlated positively with BMI, waist circumference, VAT, fasting glucose, and total cholesterol (all $p < 0.05$).

## Conclusions

Despite durable viral suppression and immune recovery, long-term ART was associated with central fat accumulation and progressive cIMT thickening, particularly at the carotid bifurcations. These findings underscore the need for cardiometabolic risk monitoring as part of routine HIV care to identify early changes that precede overt disease.

## Introduction

With the advent of combination antiretroviral therapy (ART), the life expectancy of people living with HIV (PWHIV) has significantly improved, transforming the disease into a chronic, manageable condition [1] However, this success has also led to an increased burden of non-AIDS-related comorbidities, notably cardiovascular disease (CVD), metabolic syndrome, and disorders of fat distribution. Among these, atherosclerosis and visceral or android adiposity have emerged as critical contributors to long-term morbidity and mortality in the HIV-positive population [2,3] HIV infection independently accelerates atherogenesis through mechanisms that include chronic immune activation, endothelial dysfunction, and oxidative stress [2]. Additionally, antiretroviral therapy, while essential for viral suppression and immune restoration, can induce adverse metabolic effects such as dyslipidemia, insulin resistance, and alterations in fat distribution. Rather than generalized fat loss, many patients develop increased accumulation of visceral or android fat, which is strongly associated with insulin resistance, dyslipidemia, systemic inflammation, and progression of subclinical atherosclerosis [4,5]. In contrast, gynoid fat, typically distributed in the gluteofemoral region, is thought to be metabolically protective, associated with improved lipid and glucose homeostasis and a reduced risk of cardiometabolic disease. Accordingly, the android-to-gynoid (A/G) fat ratio has emerged as a robust indicator of body fat redistribution and cardiovascular risk in both general and HIV-infected populations [6–8]. Several studies have shown that both traditional cardiovascular risk factors and HIV-related factors contribute to increased carotid intima-media thickness (cIMT), a validated surrogate marker of atherosclerosis, in this population [3,6,7,9,10]. Notably, visceral adipose tissue and a higher A/G ratio appear to significantly accelerate atherosclerosis progression, even among patients without overt fat redistribution syndromes. Moreover, while some data suggest that traditional risk factors such as hypertension, smoking, and dyslipidemia are predominant determinants

of atherosclerosis in HIV, others emphasize the additive effect of HIV-specific factors, such as chronic inflammation and prolonged ART exposure [11].

Understanding the specific contributions of these factors is crucial for developing effective strategies to mitigate the long-term health risks associated with these changes in body composition and arterial structure in the context of successful HIV treatment. This study aimed to characterize 36-month longitudinal changes in fat distribution as assessed by bioelectrical impedance analysis (BIA) and dual-energy X-ray absorptiometry (DEXA), alongside metabolic parameters and carotid intima-media thickness (cIMT), following ART initiation in a Thai cohort of PWHIV.

## Methods

### Study design and setting

This is a prospective cohort study of three-year-follow up data among people living with HIV (PWHIV) at Siriraj hospital, the largest Thailand's tertiary referral center. The objective of the study is to investigate the metabolic changes that occur in PWHIV following the initiation of antiretroviral therapy (ART).

### Study participants

This study enrolled 132 antiretroviral therapy (ART)–naïve adults (≥18 years) newly diagnosed with HIV receiving treatment at outpatient unit of Siriraj Hospital between October 2016 and December 2019. Individuals were excluded if they had pre-existing diabetes mellitus, active or uncontrolled comorbid conditions (such as cardiovascular or liver disease), were pregnant or breastfeeding, or were unable to comply with the study protocol and follow-up schedule. These exclusions were applied to minimize confounding influences on metabolic and vascular outcomes, ensuring that observed changes could be more reliably attributed to ART initiation and HIV-related factors. The study protocol was approved by the Siriraj Institutional Review Board (SIRB), Faculty of Medicine Siriraj Hospital, Mahidol University (COA no. 793/2558, approval date 15 August 2016; renewed annually through 30 May 2021). Written informed consent was obtained from all participants before enrollment. The study was conducted in accordance with the Declaration of Helsinki and local regulations. All data were de-identified prior to analysis.

Participants initiated standard first-line ART regimens in accordance with national clinical guidelines. Initial ART classes and regimens were documented, including nucleoside/nucleotide reverse transcriptase inhibitors (NRTIs), non-nucleoside reverse transcriptase inhibitors (NNRTIs), protease inhibitors (PIs), and integrase strand transfer inhibitors (INSTIs). Use of tenofovir disoproxil fumarate (TDF)-containing backbones was also specifically recorded. The distribution of initial ART classes was summarized as percentages to enable analysis of potential class-specific metabolic and vascular effects. Plasma HIV RNA and CD4 cell counts were measured at baseline and during follow-up. Viral suppression, defined as HIV RNA<40 copies/mL, was assessed at 12, 24, and 36 months post-ART initiation to confirm adherence and treatment effectiveness. These data allowed evaluation of the temporal relationship between ART exposure, immune recovery, and cardiometabolic outcomes. Concomitant use of medications for cardiovascular or metabolic risk management, such as statins, antihypertensive agents, and antidiabetic drugs, was not systematically recorded during the study. As a result, adjustment for these potential confounders could not be performed. We acknowledge this as a study limitation, as background pharmacologic interventions may have influenced lipid levels, glycemic parameters, or carotid intima-media thickness (cIMT) progression.

Follow-up assessments, including fasting blood tests, BMI measurement, body composition analysis (via BIA and DEXA), and carotid intima-media thickness (cIMT) measurement, were performed at baseline and again at 12, 24, and 36 months after ART initiation to evaluate longitudinal changes in metabolic and vascular parameters.

### Biochemical parameters

At the baseline visit, patients underwent a comprehensive assessment, including a detailed interview and structured questionnaire covering age, sociodemographic characteristics, HIV disease history, other comorbid conditions, health-related

behaviors, current medication use for hypertension, diabetes, and dyslipidemia. Following an overnight fast, blood specimens were collected for measurement of glucose, HbA1c, insulin, total cholesterol, triglycerides, HDL cholesterol, creatinine, AST and ALT. LDL cholesterol was calculated, except for patients with triglyceride of ≥400 mg/dL where LDL cholesterol was measured directly. Baseline CD4 counts and viral load measurements were also obtained. A 2-hour 75-g oral glucose tolerance test (OGTT) was administered, and HOMA-IR and HOMA-beta were calculated [12]. Estimated glomerular filtration rate (eGFR) and urine albumin-to-creatinine ratio (UACR) were also assessed.

## Body composition evaluation

Height, weight, waist circumference, and hip circumference were measured at each visit to calculate for body mass index (BMI) and waist-to-hip ratio. Body composition was assessed using bioelectrical impedance analysis (BIA; InBody 570, InBody Co., Seoul, Korea) and dual-energy X-ray absorptiometry (DEXA; GE Lunar iDXA, GE Healthcare, Madison, WI, USA). From BIA, body fat percentage, fat mass, and visceral fat level (derived from impedance-based estimation of visceral fat area) were obtained. From DEXA, total body fat, lean mass, android and gynoid fat masses, the android-to-gynoid (A/G) fat ratio, and visceral adipose tissue (VAT) were quantified. Android fat was defined as the abdominal region extending from the top of the iliac crest to 20% of the distance to the base of the skull, and gynoid fat as the hip/thigh region extending from the femoral head to mid-thigh, following manufacturer guidelines. The A/G ratio was calculated as android fat mass divided by gynoid fat mass. A higher ratio (particularly values significantly greater than 1.0) indicates disproportionate abdominal adiposity, which has been associated with increased cardiometabolic risk and adverse health outcomes [8].

## Carotid intima-media measurement

Carotid intima-media thickness (cIMT) was assessed bilaterally using B-mode ultrasound (LOGIQ E10 Series, GE Healthcare, Japan). Imaging included the common carotid artery (CCA; proximal, mid, distal, measured 3 cm below the carotid bulb), carotid bifurcation (CB), and internal carotid artery (ICA), following standardized protocols [13]. At each site, three plaque-free measurements were obtained from longitudinal images with clear lumen–intima and media–adventitia boundaries. The thickest IMT value per segment was recorded, and mean values across three measurements were averaged for analysis. Subclinical carotid atherosclerosis was defined as IMT > 0.80 mm, the presence of plaque, or both [14–16]. All measurements were performed by trained operators blinded to participants' clinical characteristics and prior scan results. Measurements were performed manually (non-automated). Intra- and inter-reader reproducibility was not formally assessed, which is acknowledged as a study limitation.

## Statistical analysis

Continuous variables were expressed as mean ± standard deviation (SD) for normally distributed data and as median with interquartile range (IQR) for skewed distributions. Categorical variables were presented as counts and percentages. Group comparisons at baseline used Student's t-test or Mann–Whitney U test for continuous variables, and the chi-square test or Fisher's exact test for categorical variables, as appropriate. For within-subject changes from baseline to the first follow-up, Wilcoxon signed-rank and McNemar's tests were applied. Longitudinal trends over 36 months were analyzed using generalized linear mixed models for repeated measures, adjusting for prespecified covariates including age, baseline fasting glucose, triglycerides, systolic blood pressure, and waist-to-hip ratio. Correlation analyses were performed using Spearman's rank test to assess associations between cIMT (bilateral mean of segment-specific values, unless otherwise specified) and clinical/metabolic parameters. To examine associations between ART drug class exposure (NRTIs, NNRTIs, PIs, INSTIs, CCR5 inhibitors) and cIMT at the right carotid bifurcation, Kendall's tau-b correlation was used, which is appropriate for ordinal/categorical exposures with non-parametric distributions. Correlation coefficients (τ) and two-tailed p values were reported. *p values* were adjusted and rounded to three decimal places; values reported as

*p = 1.000* reflect adjusted probabilities ≥0.99. Analyses were performed in SPSS v24.0. All tests were two-sided (α = 0.05). Distributions were assessed and transformations applied as needed. Outliers were examined with influence diagnostics; analyses were repeated with and without influential points. Longitudinal trends were assessed via generalized linear mixed models with random intercepts (unstructured covariance), adjusted for prespecified covariates (age, baseline fasting glucose, triglycerides, systolic blood pressure, waist-to-hip ratio). We report effect sizes (β) with 95% CIs and p-values.

## Results

### Baseline characteristics and ART exposure

A total of 132 people living with HIV (PWHIV) were enrolled at baseline. The mean age was 30.9 ± 8.1 years; most participants were male (n = 116, 87.9%), and heterosexual orientation was most common (69.3%). The mean baseline CD4 count was 451.5 ± 179.3 cells/μL, and the mean baseline HIV RNA level was 4.4 ± 0.8 log10 copies/mL. All participants initiated antiretroviral therapy (ART) at enrollment. Table 1 summarizes exposure to major ART classes during the 36-month follow-up. Nucleoside reverse transcriptase inhibitors (NRTIs) were prescribed to all participants, with 36.4% (n = 48) remaining on the same regimen and 63.6% (n = 84) switching regimens during follow-up. Non-nucleoside reverse transcriptase inhibitors (NNRTIs) were used by 94.0% overall; 28.0% (n = 37) continued therapy throughout, 65.9% (n = 87) discontinued, and 6.1% (n = 8) were never exposed. Protease inhibitors (PIs) were rarely prescribed (5.3%, n = 7); 2.3% (n = 3) remained on therapy, 3.0% (n = 4) discontinued, and 94.7% (n = 125) were never exposed. Integrase strand transfer inhibitors (INSTIs) were received by 65.2% overall; 32.6% (n = 43) continued therapy throughout, 2.3% (n = 3) discontinued, and 65.2% (n = 86) never received an INSTI. No participants were prescribed CCR5 inhibitors.

### Virological suppression and CD4 immune recovery

By 12 months after ART initiation, 82.0% of participants achieved virological suppression (HIV RNA < 40 copies/mL), increasing to 91.0% at 24 months and 92.2% at 36 months, consistent with sustained treatment exposure and adherence. Median CD4 count rose steadily from 410 cells/μL (IQR 322–566) at baseline to 643 cells/μL (IQR 496–776) at 12 months, 649 cells/μL (IQR 524–778) at 24 months, and 716 cells/μL (IQR 563–854) at 36 months, reflecting progressive immune reconstitution. Virological suppression rates and CD4 trajectories are summarized in Table 2, with CD4 comparisons across timepoints evaluated using the Kruskal–Wallis test and pairwise contrasts with baseline assessed by the Mann–Whitney U test (all p < 0.0001).

**Table 1. Exposure to major antiretroviral therapy (ART) classes during 36 months of follow-up.**

| ART class | Remained throughout n (%) | Discontinued n (%) | Never received n (%) |
|---|---|---|---|
| NRTIs | 48 (36.4) | 84 (63.6) | 0 (0) |
| NNRTIs | 37 (28.0) | 87 (65.9) | 8 (6.1) |
| PIs | 3 (2.3) | 4 (3.0) | 125 (94.7) |
| INSTIs | 43 (32.6) | 3 (2.3) | 86 (65.2) |
| CCR5 | 0 (0) | 0 (0) | 132 (100) |

Distribution of exposure to major ART drug classes.

Values are presented as number (%) of participants. "Remained throughout" indicates continuous exposure during all study visits; "discontinued" indicates exposure at some point but not maintained through 36 months; "never received" indicates no exposure to that drug class. NRTI, nucleoside reverse transcriptase inhibitor; NNRTI, non-nucleoside reverse transcriptase inhibitor; PI, protease inhibitor; INSTI, integrase strand transfer inhibitor; CCR5i, CCR5 inhibitor.

**Table 2. Virological suppression and CD4 immune recovery over 36 months of ART.**

| Timepoint | Participants assessed (n) | Virological suppression* n (%) | Median CD4 count (cells/µL) | IQR (25th–75th percentile) | p-value vs baseline† |
|---|---|---|---|---|---|
| Baseline | 132 | 0 (0.0) | 410 | 322–566 | – |
| 12 months | 128 | 105 (82.0) | 643 | 496–776 | <0.0001 |
| 24 months | 111 | 101 (91.0) | 649 | 524–778 | <0.0001 |
| 36 months | 103 | 95 (92.2) | 716 | 563–854 | <0.0001 |

*Virological suppression defined as HIV RNA<40 copies/mL. CD4 counts are reported as median with interquartile range (IQR).

†p-values calculated using Mann–Whitney U test comparing each follow-up to baseline. Overall Kruskal–Wallis test across timepoints: $\chi^2 = 53.6$, $p < 0.0001$.

### Anthropometric measurement

At baseline, anthropometric assessments showed a mean body weight of 65.9 ± 13.8 kg and a mean body mass index (BMI) of 23.0 ± 4.5 kg/m², with BMI values ranging from 14.8 to 38.3 kg/m², indicating the presence of both underweight and overweight individuals within the cohort. Central adiposity was evident, with an average waist circumference of 82.5 ± 11.4 cm and hip circumference of 91.7 ± 8.9 cm. Among HIV-infected participants receiving ART, BMI increased significantly over time, rising by 2.2% at 24 months ($p < 0.05$) and 3.0% at 36 months ($p < 0.01$). This was accompanied with a parallel and statistically significant increase in body weight, waist circumference, and hip circumference over the same period ($p < 0.01$ for all). These findings indicate a progressive accumulation of central adiposity associated with long-term ART exposure (Table 3).

### Body composition and fat distribution

Body composition assessment using bioelectrical impedance analysis (BIA) demonstrated a mean baseline body fat percentage of 22.3 ± 0.8%. Significant increases in total body fat were observed, rising by 5.8% at 24 months ($p < 0.01$) and 6.7% at 36 months ($p < 0.05$), reaching levels consistent with body fat component (Table 3).

Complementary DEXA scans showed a significant increase in android fat without a corresponding change in gynoid fat, resulting in a marked rise in the android-to-gynoid (A/G) fat ratio at both 24 months ($p < 0.05$) and 36 months ($p < 0.01$) following the initiation of ART. Notably, the A/G ratio exceeded 1.0 after 24 months, indicating a shift toward central fat accumulation. Concurrently, visceral adipose tissue (VAT) mass increased significantly from a baseline of 518.7 ± 41.4 g to 622.0 ± 47.5 (19.9%) and 689.7 ± 51.2 (33.0%) higher levels at 24 and 36 months, respectively ($p < 0.01$) (Table 3).

### Metabolic parameters

Significant metabolic alterations were observed during the follow-up period following the initiation of ART. Fasting glucose levels increased significantly over time ($p < 0.05$). Additionally, HOMA-Beta, an index of pancreatic beta-cell function, showed a significant decline at 24 months ($p < 0.05$), indicating a temporary reduction in insulin secretory capacity at that time point (Table 3).

There were no significant changes observed in HbA1c, total cholesterol, HDL, triglycerides, LDL, HOMA-IR, or the prevalence of metabolic syndrome throughout the study period. Systolic blood pressure showed a significant decreasing trend ($p < 0.01$), while diastolic blood pressure remained stable (Table 3).

Renal function, assessed by estimated glomerular filtration rate (eGFR), showed a significant decline over time ($p < 0.05$) after ART initiation. However, albuminuria did not change significantly. These findings suggest that long-term ART may contribute to gradual metabolic dysregulation and a decline in renal function (Table 3).

**Table 3. Anthropometric, body composition, metabolic, and renal parameters during 36 months of ART.**

| | Baseline | 12 month | P-value | 24 month | P-value | 36 month | P-value |
|---|---|---|---|---|---|---|---|
| **Body mass index (BMI), kg/m²** | 23.03±0.39 | 23.13±0.42 | 1.000 | 23.54±0.41 | <0.05 | 23.73±0.43 | < 0.01 |
| **Waist circumference, cm** | 82.32±1.01 | 83.40±1.01 | 0.124 | 84.66±1.01 | <0.01 | 84.99±1.06 | < 0.01 |
| **Hip circumference, cm** | 91.29±0.79 | 92.58±0.81 | <0.05 | 93.81±0.81 | <0.01 | 95.05±0.83 | < 0.01 |
| **SBP, mmHg** | 121.0±11.2 | 119.0±10.6 | <0.05 | 117.9±11.2 | <0.01 | 118.3±11.7 | < 0.01 |
| **DBP, mmHg** | 72.7±9.1 | 72.6±8.8 | 1.000 | 71.1±7.7 | 0.068 | 72.6±9.8 | 1.000 |
| **Body composition analysis by BIA** | | | | | | | |
| %  Body fat | 22.27±9.38 | 23.28±9.53 | 0.745 | 24.38±9.56 | <0.01 | 24.91±9.47 | < 0.05 |
| Fat mass, kg | 15.33±8.84 | 16.09±9.13 | 0.748 | 17.22±9.32 | < 0.05 | 18.13±10.42 | 0.106 |
| **Body compostion analysis by DEXA** | | | | | | | |
| %  Android fat | 27.89±12.20 | 29.23±12.40 | 0.509 | 31.05±12.61 | <0.05 | 32.83±12.95 | < 0.01 |
| %  Gynoid fat | 28.03±8.57 | 28.63±8.84 | 0.765 | 28.98±8.09 | 0.322 | 29.49±8.78 | 0.115 |
| **A/G ratio** | 0.97±0.26 | 0.99±0.25 | 1.000 | 1.01±0.24 | < 0.05 | 1.03±0.25 | < 0.01 |
| **Visceral adipose tissue mass, g** | 518.7±41.4 | 550.2±43.6 | 1.000 | 622.0±47.5 | < 0.01 | 689.7±51.2 | < 0.01 |
| **FBS, mg/dL** | 85.51±11.90 | 88.66±11.20 | <0.01 | 88.95±20.10 | 0.188 | 89.35±16.27 | < 0.05 |
| **HbA1C, %** | 5.36±0.45 | 5.30±0.38 | 0.120 | 5.34±0.50 | 1.000 | 5.35±0.60 | 1.000 |
| **Total cholesterol, mg/dL** | 169.42±31.22 | 169.49±30.55 | 1.000 | 171.58±32.02 | 1.000 | 175.50±34.37 | 0.350 |
| **Triglyceride, mg/dL** | 110.39±51.05 | 118.29±66.89 | 1.000 | 125.46±76.59 | 0.217 | 120.94±67.01 | 0.901 |
| **HDL, mg/dL** | 46.10±11.25 | 47.63±9.80 | 0.615 | 47.71±10.53 | 0.824 | 48.24±10.91 | 0.379 |
| **c-LDL, mg/dL** | 101.24±27.33 | 98.14±25.13 | 0.951 | 99.25±28.05 | 1.000 | 102.41±30.14 | 1.000 |
| **eGFR, mL/min/1.73m²** | 109.61±12.01 | 107.62±13.55 | 0.378 | 106.65±14.93 | 0.134 | 103.85±14.57 | < 0.01 |
| **UACR, mg/g Cr** | 18.40±6.40 | 30.19±16.22 | 1.000 | 26.44±12.93 | 1.000 | 20.64±8.63 | 1.000 |
| **HOMA-IR** | 2.66±0.32 | 2.61±0.23 | 1.000 | 2.48±0.25 | 1.000 | 2.84±0.29 | 1.000 |
| **HOMA-Beta** | 208.85±17.79 | 171.64±11.43 | 0.106 | 165.73±10.01 | 0.049 | 183.19±14.31 | 0.635 |

Longitudinal changes in anthropometry, body fat distribution, glycemic indices, lipid profiles, and renal function.

Values are presented as mean±SD, unless otherwise specified.

*P-values* represent comparisons with baseline values at each timepoint.

Abbreviations: BMI = body mass index; SBP = systolic blood pressure; DBP = diastolic blood pressure; BIA = bioelectrical impedance analysis; DEXA = dual-energy X-ray absorptiometry; FBS = fasting blood sugar; HbA1c = glycated hemoglobin; HDL = high-density lipoprotein; c-LDL = calculated low-density lipoprotein; eGFR = estimated glomerular filtration rate; UACR = urinary albumin-to-creatinine ratio; HOMA-IR = homeostasis model assessment of insulin resistance; HOMA-β = homeostasis model assessment of beta-cell function.

## Carotid intima-media thickness (cIMT)

Ultrasound assessments demonstrated progressive increases in cIMT across multiple arterial sites following ART initiation. Significant cIMT progression was observed at both the right and left carotid bifurcations during the 36-month follow-up. At the left carotid bifurcation, cIMT increased from 0.675±0.016 mm at baseline to 0.745±0.021 mm at 24 months and 0.756±0.023 mm at 36 months (p<0.01). At the right carotid bifurcation, cIMT rose from 0.644±0.013 mm at baseline to 0.686±0.015 mm at 12 months (p<0.05), 0.716±0.023 mm at 24 months (p<0.05), and 0.729±0.019 mm at 36 months (p<0.01). In addition to bifurcation sites, upward trends were also observed in both the common carotid artery (CCA) and the internal carotid artery (ICA), bilaterally (Table 4), suggesting gradual and widespread progression of subclinical atherosclerosis during long-term ART. The longitudinal changes at both carotid bifurcation sites are illustrated in Fig 1, highlighting the statistically significant progression of cIMT over the 36-month follow-up.

Because the right carotid bifurcation demonstrated the most consistent and statistically significant progression compared with other arterial sites, subsequent correlation analyses were performed using changes in cIMT at this site as the primary outcome. As shown in Table 5, increases in right carotid bifurcation cIMT were significantly and positively

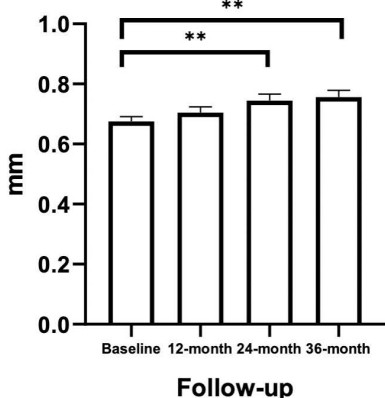

**Table 4. Progression of carotid intimal media thickness (cIMT) after ART treatment.**

| cIMT, cm | Baseline | 12 month | P-value | 24 month | P-value | 36 month | P-value |
|---|---|---|---|---|---|---|---|
| **Carotid bifurcation (CB)** | | | | | | | |
| Right | 0.644±0.013 | 0.686±0.015 | < 0.05 | 0.716±0.023 | < 0.05 | 0.729±0.019 | < 0.01 |
| Left | 0.675±0.016 | 0.705±0.019 | 0.704 | 0.745±0.021 | <0.01 | 0.756±0.023 | < 0.01 |
| **Proximal carotid artery (PCCA)** | | | | | | | |
| Right | 0.582±0.006 | 0.598±0.007 | 0.182 | 0.592±0.008 | 1.000 | 0.605±0.008 | 0.056 |
| Left | 0.559±0.007 | 0.564±0.007 | 1.000 | 0.566±0.009 | 1.000 | 0.570±0.008 | 1.000 |
| **Distal common carotid artery (DCCA)** | | | | | | | |
| Right | 0.586±0.010 | 0.584±0.010 | 1.000 | 0.601±0.011 | 1.000 | 0.590±0.012 | 1.000 |
| Left | 0.584±0.010 | 0.585±0.008 | 1.000 | 0.595±0.010 | 1.000 | 0.594±0.010 | 1.000 |
| **Mid common carotid artery (MCCA)** | | | | | | | |
| Right | 0.544±0.008 | 0.545±0.008 | 1.000 | 0.546±0.010 | 1.000 | 0.561±0.009 | 0.582 |
| Left | 0.542±0.008 | 0.552±0.008 | 1.000 | 0.559±0.008 | 0.198 | 0.574±0.010 | < 0.01 |
| **Internal carotid artery (ICA)** | | | | | | | |
| Right | 0.607±0.013 | 0603±0.014 | 1.000 | 0.624±0.013 | 1.000 | 0.620±0.016 | 1.000 |
| Left | 0.607±0.011 | 0.595±0.016 | 1.000 | 0.628±0.015 | 1.000 | 0.654±0.016 | < 0.05 |

Segment-specific cIMT measurements at baseline, 12, 24, and 36 months.

*P-values* represent comparisons with baseline values.

cIMT was measured bilaterally at the common carotid artery (proximal, mid, distal), carotid bifurcation, and internal carotid artery.

Abbreviations: LCB=left carotid bifurcation; LPCCA=left proximal common carotid artery; LDCCA=left distal common carotid artery; LMCCA=left mid common carotid artery; LICA=left internal carotid artery; RCB=right carotid bifurcation; RPCCA=right proximal common carotid artery; RDCCA=right distal common carotid artery; RMCCA=right mid common carotid artery; RICA=right internal carotid artery.

**Fig 1. Progression of carotid intima-media thickness (cIMT) at the right and left carotid bifurcations during 36 months of ART.** (A) Right carotid bifurcation cIMT at baseline, 12, 24, and 36 months after ART initiation (mean±SEM). (B) Left carotid bifurcation cIMT at the same time points (mean±SEM). Both right and left carotid bifurcations demonstrated significant increases in cIMT during 36 months of ART. The right bifurcation showed earlier and more consistent progression, with significant increases already at 12 months, while the left bifurcation exhibited significant increases from 24 months onward. Error bars represent SEM. *p<0.05, **p<0.01 versus baseline. Abbreviations: ART, antiretroviral therapy; cIMT, carotid intima–media thickness; SEM, standard error of the mean; CB, carotid bifurcation.

**Table 5. Correlations between carotid intima-media thickness (cIMT) of right carotid bifurcation and body composition/metabolic parameters.**

| Correlations between cIMT and: | R | P-value |
|---|---|---|
| **Body mass index (BMI), kg/m²** | 0.316 | *<0.001* |
| **Waist circumference, cm** | 0.279 | *<0.001* |
| **Hip circumference, cm** | 0.294 | *<0.001* |
| **SBP, mmHg** | 0.122 | *0.016* |
| **DBP, mmHg** | 0.169 | *0.001* |
| **Body composition analysis by BIA** | | |
| % Body fat | 0.203 | *<0.001* |
| Fat mass, kg | 0.237 | *<0.001* |
| **Body compostion analysis by DEXA** | | |
| % Android fat | 0.247 | *<0.001* |
| % Gynoid fat | 0.126 | *0.014* |
| **A/G ratio** | 0.277 | *<0.001* |
| **Visceral adipose tissue mass, g** | 0.288 | *<0.001* |
| **FBS, mg/dL** | 0.167 | *0.001* |
| **HbA1C, %** | 0.075 | *0.140* |
| **Total cholesterol, mg/dL** | 0.108 | *0.034* |
| **Triglyceride, mg/dL** | 0.080 | *0.115* |
| **HDL, mg/dL** | 0.018 | *0.728* |
| **c-LDL, mg/dL** | 0.078 | *0.125* |
| **eGFR, mL/min/1.73m²** | −0.145 | *0.004* |
| **UACR, mg/g Cr** | 0.053 | *0.299* |
| **HOMA-IR** | 0.038 | *0.452* |
| **HOMA-Beta** | −0.071 | *0.163* |

Associations of anthropometric, body fat, and metabolic measures with cIMT progression of right carotid bifucation.

Correlation coefficients (*r*) are Spearman values.

*P*-values <0.05 are considered statistically significant.

cIMT values used for correlation represent the mean of bilateral maximum measurements across all segments.

Abbreviations: BMI = body mass index; SBP = systolic blood pressure; DBP = diastolic blood pressure; VAT = visceral adipose tissue; FBS = fasting blood sugar; HbA1c = glycated hemoglobin; HDL = high-density lipoprotein; c-LDL = calculated low-density lipoprotein; eGFR = estimated glomerular filtration rate; UACR = urinary albumin-to-creatinine ratio; HOMA-IR = homeostasis model assessment of insulin resistance; HOMA-β = homeostasis model assessment of beta-cell function.

correlated with anthropometric and metabolic parameters, including BMI, waist and hip circumferences, systolic and diastolic blood pressure, fat mass, total body fat percentage, percent android fat, percent gynoid fat, android-to-gynoid fat ratio, visceral adipose tissue (VAT) mass, fasting blood sugar (FBS), and total cholesterol. In contrast, no significant correlations were observed with HbA1c, triglycerides, HDL, calculated LDL cholesterol (c-LDL), urinary albumin-to-creatinine ratio (UACR), HOMA-IR, or HOMA-β.

## Correlation of RcIMT with ART drug class

When analyses were adjusted for ART drug exposure, right carotid bifurcation cIMT demonstrated a negative correlation with NNRTI use ($\tau = -0.085$, $p = 0.038$) and with protease inhibitor use ($\tau = -0.109$, $p = 0.009$). No significant correlations

were found with NRTIs (τ = 0.047, p = 0.264), INSTIs (τ = 0.014, p = 0.735), or CCR5 inhibitors (not prescribed in this cohort). Right carotid bifurcation cIMT was also not associated with baseline viral load (τ = −0.022, p = 0.532) or baseline CD4 count (τ = −0.012, p = 0.720) (Table 6).

## Discussion

This prospective cohort study provides longitudinal evidence on the impact of combination antiretroviral therapy (ART) on body composition, metabolic parameters, and subclinical atherosclerosis in Thai adults living with HIV. Over 36 months of follow-up, we observed progressive increases in visceral adiposity and carotid intima-media thickness (cIMT), underscoring the complex interplay between ART exposure, fat redistribution, and early cardiovascular risk.

Consistent with previous reports, ART initiation in our cohort was associated with significant gains in body fat, particularly central adiposity [17,18]. The concurrent increases in BMI, waist and hip circumference, and total body fat percentage support the notion that long-term ART promotes unfavorable fat redistribution [19]. Notably, DEXA scans demonstrated a marked increase in android fat without parallel changes in gynoid fat, resulting in a sustained elevation of the android-to-gynoid (A/G) ratio. The A/G ratio exceeding 1.0 after 24 months, together with significant gains in visceral adipose tissue (VAT) mass, underscores the development of central lipohypertrophy, a hallmark of HIV-associated adipose tissue redistribution [3,18]. At the same time, part of the observed weight and fat gain may reflect the "return-to-health" phenomenon, particularly among participants who were underweight or had low baseline BMI prior to ART initiation. Recovery of lean and fat mass in this subgroup likely represents a favorable response to viral suppression and immune reconstitution, rather than purely pathological fat accumulation. Nevertheless, the disproportionate and sustained central fat gain documented by DEXA and VAT measures suggests that, beyond physiologic catch-up, ART contributes to redistribution patterns with potential cardiometabolic implications [3,20].

This redistribution of fat has clinical importance given its strong association with metabolic dysfunction. In our study, early metabolic perturbations were observed, including an increase in fasting plasma glucose and a transient decline in HOMA-β at 24 months, suggesting a temporary reduction in pancreatic β-cell secretory capacity. HOMA-IR, however, remained relatively stable throughout follow-up, indicating no major sustained change in insulin resistance during this period. Although HbA1c and lipid profiles also remained stable, these subtle metabolic shifts highlight the importance of early and ongoing monitoring. Similar transient disturbances in β-cell indices have been reported in other ART-treated cohorts, where early

**Table 6. Correlation between right carotid bifurcation cIMT and ART drug class exposure.**

| ble | Kendall's τ-b | *p*-value |
| --- | --- | --- |
| NRTIs (exposure) | 0.047 | *0.264* |
| NNRTIs (exposure) | −0.085 | *0.038 ** |
| PIs (exposure) | −0.109 | *0.009 **** |
| INSTIs (exposure) | 0.014 | *0.735* |
| CCR5 inhibitors | – | – |
| Baseline viral load | −0.022 | *0.532* |
| Baseline CD4 count | −0.012 | *0.720* |

Kendall's tau-b correlation coefficients are reported.

Negative values indicate an inverse correlation between RcIMT and exposure.

*p* < 0.05 considered significant (*p* < 0.05 = *, *p* < 0.01 = **).

CCR5 inhibitors were not prescribed in this cohort.

Abbreviations: cIMT = carotid intima-media thickness; RcIMT = right carotid bifurcation cIMT; NRTIs = nucleoside reverse transcriptase inhibitors; NNRTIs = non-nucleoside reverse transcriptase inhibitors; PIs = protease inhibitors; INSTIs = integrase strand transfer inhibitors; CCR5 = CCR5 receptor antagonists.

fluctuations did not always translate into persistent dysfunction, but may still signal long-term metabolic vulnerability [21]. Interestingly, the prevalence of metabolic syndrome did not significantly increase, which may reflect the cohort's relatively young age and preserved immune function at baseline. In contrast, declines in estimated glomerular filtration rate (eGFR) raise concerns that renal function may be more vulnerable to long-term ART effects, even in the absence of albuminuria.

Interpretation of these outcomes requires consideration of regimen distribution. Nearly all participants received NRTIs and most were treated with NNRTIs, while only one-third were exposed to INSTIs and few to protease inhibitors (PIs). No participants received CCR5 inhibitors, reflecting Thailand's national guidelines during the study period. Because ART-associated metabolic and renal effects vary by drug class, the predominance of NNRTI-based regimens and limited PI or INSTI exposure may have influenced the observed outcomes. For example, modest eGFR declines may be partly attributable to tenofovir disoproxil fumarate or transporter-mediated effects of INSTIs. Similarly, the relative stability of lipid profiles may relate to the small number of participants exposed to PIs, which are more strongly linked to dyslipidemia and insulin resistance. Thus, while overall trends are clear, regimen-specific effects could not be fully disentangled in this cohort. Larger, stratified studies with longer follow-up are needed.

The most striking vascular finding was the progressive increase in cIMT across multiple arterial segments, particularly at the carotid bifurcations. Because the right carotid bulb showed the earliest and most consistent progression compared with other arterial sites, we used this segment for correlation analyses. Right carotid bulb cIMT was positively correlated with BMI, waist and hip circumference, total and visceral fat mass, and A/G ratio, as well as fasting blood sugar and total cholesterol. These associations support a mechanistic link between fat redistribution, metabolic dysregulation, and vascular remodeling in people with HIV on ART [2,17,22]. By contrast, no associations were observed with HbA1c, triglycerides, HDL, calculated LDL, UACR, HOMA-IR, or HOMA-β, suggesting that certain traditional markers may underestimate early vascular injury in this population.

Collectively, these findings emphasize the need for HIV care strategies that extend beyond viral suppression. Early identification of individuals at risk of fat redistribution, metabolic derangement, and vascular injury is essential. Tailored interventions—including lifestyle modification, ART regimen optimization, and timely use of cardioprotective therapies—may mitigate long-term cardiovascular risk [2,23,24].

Our study also highlights the value of integrating body composition and vascular imaging into HIV follow-up. Serial BIA, DEXA, and carotid ultrasound provided complementary insights into fat redistribution and subclinical atherosclerosis over time. Importantly, BIA-derived fat mass, which is non-invasive and accessible in most clinical settings, may serve as a practical early marker for cardiovascular risk before biochemical abnormalities emerge.

## Limitations

This study has limitations. It was conducted at a single tertiary center with a modest sample size, which may limit generalizability. The 36-month follow-up, while sufficient to detect significant changes, may not capture the long-term trajectory of metabolic and vascular complications associated with lifelong ART. Although ART regimens were recorded, analyses were not stratified by drug class, limiting evaluation of regimen-specific effects. This is important given the known metabolic and renal effects of certain agents, and the potential influence of transporter interactions (e.g., integrase inhibitors, cobicistat) on creatinine and eGFR, which may represent pseudo-declines rather than true impairment. The absence of a control group also prevents separation of ART-related changes from those attributable to natural aging or return-to-health after ART initiation. Carotid IMT was measured manually rather than by automated edge detection, so very small changes should be interpreted cautiously. Finally, concomitant use of cardiometabolic medications such as statins, antihypertensives, or antidiabetics and sociodemographic characteristics were not systematically captured, which may have influenced outcomes. However, use of these drugs could retard metabolic derangement. For example, statins are effective at decreasing or stabilizing IMT. If some patients in this study were taking statins and their IMT still increased, it suggests that the effect of ART on IMT was stronger than the effect of the statins.

Nevertheless, this study also has important strengths. To our knowledge, it is among the first prospective cohorts in Southeast Asia to provide a comprehensive evaluation of ART-associated changes in body composition, metabolic health, and vascular function. The use of multimodal assessments, including BIA for accessible fat mass screening, DEXA for precise regional adiposity quantification, and serial carotid ultrasound for subclinical atherosclerosis, allowed for a robust and integrated analysis of cardiometabolic risk. Furthermore, the 36-month longitudinal follow-up with repeated measures strengthened our ability to detect progressive changes over time, yielding novel insights into the dynamic interplay between ART, fat redistribution, and vascular remodeling in people living with HIV.

## Conclusion

In summary, this longitudinal study demonstrates that long-term ART in HIV-positive individuals is associated with progressive visceral adiposity, subtle metabolic dysregulation, and measurable progression of carotid atherosclerosis, particularly at the carotid bifurcation. These findings reinforce the need for integrated HIV care models that incorporate cardiometabolic monitoring and risk reduction alongside viral suppression. Proactive strategies addressing fat redistribution, metabolic health, and vascular integrity may be essential to improving long-term outcomes in people living with HIV.

## Supporting information

**S1 Table. Baseline characteristics of study participants (n = 132).** Demographic and clinical characteristics at enrollment, including age, sex, anthropometric indices (height, weight, waist and hip circumference), fasting glucose, HbA1c, lipid profile, and renal function measures. Values are presented as individual-level de-identified data. Data are provided as a CSV file and archived in Figshare (DOI: 10.6084/m9.figshare.30032515).
(XLSX)

**S2 Table. Anthropometric, metabolic, and renal parameters during 36 months of ART.** Summary statistics of longitudinal changes in body weight, waist and hip circumference, fasting plasma glucose, HbA1c, lipid profile (total cholesterol, triglycerides, HDL, LDL), serum creatinine, and estimated glomerular filtration rate (eGFR). Data are presented at baseline, 12, 24, and 36 months after ART initiation, and include mean ± SD, median, minimum, maximum, and sample size at each timepoint. Data are provided as a CSV file and archived in Figshare (DOI: 10.6084/m9.figshare.30032515).
(XLSX)

**S3 Table. Body composition assessed by BIA and DXA during 36 months of ART.** Summary of longitudinal changes in body composition parameters measured by bioelectrical impedance analysis (BIA) and dual-energy X-ray absorptiometry (DXA). Variables include fat mass (kg), body fat percentage, visceral adipose tissue (VAT, kg) by BIA, and fat mass (kg), body fat percentage, and android-to-gynoid fat ratio by DXA. Data are presented at baseline, 12, 24, and 36 months after ART initiation, and include mean ± SD, median, minimum, maximum, and sample size at each timepoint. Data are provided as a CSV file and archived in Figshare (DOI: 10.6084/m9.figshare.30032515).
(XLSX)

**S4 Table. Carotid intima-media thickness (IMT) measurements during 36 months of ART.** Summary of carotid IMT values assessed by ultrasound at baseline, 12, 24, and 36 months after ART initiation. Variables include common carotid artery (CCA) IMT at proximal, mid, and distal segments (right and left), bifurcation IMT, and internal carotid artery (ICA) IMT. Data are presented as mean ± SD, median, minimum, maximum, and sample size at each timepoint. Data are provided as a CSV file and archived in Figshare (DOI: 10.6084/m9.figshare.30032515).
(XLSX)

**S5 Table. Immunological and virological outcomes over 36 months of ART.** Median (interquartile range) CD4 cell counts and the percentage of participants achieving virological suppression (HIV RNA < 40 copies/mL) are reported at

baseline, 12, 24, and 36 months following ART initiation. Data are provided as a CSV file and archived in Figshare (DOI: 10.6084/m9.figshare.30032515).
(XLSX)

**S6 Table. Antiretroviral drug class exposure during 36 months of follow-up.** Distribution of participants exposed to each major ART class, including nucleoside/nucleotide reverse transcriptase inhibitors (NRTIs), non-nucleoside reverse transcriptase inhibitors (NNRTIs), protease inhibitors, integrase strand transfer inhibitors, and CCR5 inhibitors. Exposure is categorized as remained on therapy throughout, discontinued, or never exposed during the 36-month follow-up period. Data are provided as a CSV file and archived in Figshare (DOI: 10.6084/m9.figshare.30032515).
(XLSX)

**S7 Table. Longitudinal metabolic parameters over 36 months of ART.** Individual-level data on anthropometric, hemodynamic, and biochemical variables collected at baseline and at 12, 24, and 36 months of antiretroviral therapy. Parameters include age, sex, body mass index (BMI), waist and hip circumference, blood pressure, fasting glucose, HbA1c, insulin, lipid profile (total cholesterol, triglycerides, HDL-C, LDL-C), liver enzymes (AST, ALT), and urinary albumin-to-creatinine ratio. Data are provided as a CSV file and archived in Figshare (DOI: 10.6084/m9.figshare.30032515).
(XLSX)

**S8 Table. Longitudinal body composition over 36 months of ART.** Individual-level data from bioelectrical impedance analysis (BIA) and dual-energy X-ray absorptiometry (DEXA) performed at baseline and at 12, 24, and 36 months of antiretroviral therapy. Parameters include fat mass and percentage body fat (BIA), visceral adipose tissue (VAT), android and gynoid fat percentages, and android-to-gynoid fat ratio. Data are provided as a CSV file and archived in Figshare (DOI: 10.6084/m9.figshare.30032515).
(XLSX)

**S9 Table. Carotid intima-media thickness (IMT) over 36 months of ART.** Individual-level ultrasound measurements of carotid intima-media thickness collected at baseline and at 12, 24, and 36 months of antiretroviral therapy. Parameters include right and left common carotid artery (proximal, mid, distal), carotid bifurcation, and internal carotid artery segments. Data are provided as a CSV file and archived in Figshare (DOI: 10.6084/m9.figshare.30032515).
(XLSX)

## Author contributions

**Conceptualization:** Thirajit Boonsaen.

**Data curation:** Thirajit Boonsaen, Phatharajit Phatharodom, Oranich Navanukroh.

**Formal analysis:** Thirajit Boonsaen.

**Investigation:** Thirajit Boonsaen, Winai Ratanasuwan, Boonrat Tassaneetrithep, Tanyaluck Thientunyakit, Jitsupa Wongsripuemtet, Shanigarn Thiravit, Mayuree Homsanit.

**Methodology:** Thirajit Boonsaen.

**Project administration:** Mayuree Homsanit.

**Supervision:** Winai Ratanasuwan, Mayuree Homsanit.

**Writing – original draft:** Thirajit Boonsaen.

**Writing – review & editing:** Thirajit Boonsaen, Mayuree Homsanit.

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
