## [Decision Letter · Decision Letter 0]

11 Jul 2025

Dear Dr. Boonsaen,

Thank you for submitting your manuscript to PLOS ONE. After careful consideration, we feel that it has merit but does not fully meet PLOS ONE’s publication criteria as it currently stands. Therefore, we invite you to submit a revised version of the manuscript that addresses the points raised during the review process.

We look forward to receiving your revised manuscript.

Kind regards,

Giuseppe Vittorio De Socio, MD, PhD

Academic Editor

PLOS ONE

Journal Requirements:

2. Thank you for stating the following financial disclosure: [the Research Development Grant, Faculty of Medicine Siriraj Hospital, Mahidol University. Research Project Code (IO): R016034002]. 

3.Please update your submission to use the PLOS LaTeX template. The template and more information on our requirements for LaTeX submissions can be found at http://journals.plos.org/plosone/s/latex .

4.Please include your full ethics statement in the ‘Methods’ section of your manuscript file. In your statement, please include the full name of the IRB or ethics committee who approved or waived your study, as well as whether or not you obtained informed written or verbal consent. If consent was waived for your study, please include this information in your statement as well.

Reviewers' comments:

Reviewer's Responses to Questions

**Comments to the Author**

1. Is the manuscript technically sound, and do the data support the conclusions?

Reviewer #1: Partly

Reviewer #2: Partly

2. Has the statistical analysis been performed appropriately and rigorously?

Reviewer #1: I Don't Know

Reviewer #2: No

3. Have the authors made all data underlying the findings in their manuscript fully available?

Reviewer #1: No

Reviewer #2: Yes

4. Is the manuscript presented in an intelligible fashion and written in standard English?

Reviewer #1: Yes

Reviewer #2: Yes

Reviewer #1: In this study, Thirajit Boonsaen and colleagues aim to investigate metabolic changes occurring during the first 36 months of antiretroviral therapy. They prospectively observed a cohort of 132 ART-naïve adults, documenting changes in blood parameters (glucose, cholesterol, creatinine and insulin resistance), body mass index (BMI), waist and hip circumference, fat percentage, and visceral adipose tissue (VAT), as measured by dual-energy X-ray absorptiometry (DEXA), as well as intima-media thickness (cIMT), as measured by B-mode ultrasound. Significant weight and fat gain was observed over the 36-month study period, alongside central fat accumulation and increases in cIMT values, without any clinically significant alterations in the analysed blood parameters. The strengths of this study are the prospective observation and the availability of different measures of metabolic health at 12, 24 and 36 months, including blood parameters, DEXA scans and cIMT evaluations. However, there are also many limitations which are not adequately addressed or discussed.

The main limitation of the study is that the type of antiretroviral therapy used by the study population is neither mentioned nor associated with the study results, which could be strongly influenced by the type of drug used (see, for example, the relationship between lipid levels and the use of protease inhibitors, or creatinine levels and the use of anchor drugs that inhibit renal transporters). Furthermore, the study population appears to be highly heterogeneous, combining advanced and recent infections, men and women, and underweight and overweight individuals. Also, the lack of virological data during follow-up could be limiting, as the authors attribute many of the changes observed during the study to ART exposure, but virological suppression (and thus presumed adherence to ART) is never evaluated. Also, there is no control group available, and we cannot rule out the possibility that the same changes would be observed in an ART-free context or in individuals undergoing chronic treatment, with analogue observation of three years' ageing.

Finally, although the changes observed in the study are statistically significant, they are never compared with a measure that could indicate the clinical impact of such changes (by comparing the observed values with recognised clinically significant cut-offs).

Here are my comments:

ASTRACT

Combination antiretroviral therapy (cART) and (PLHIV) . Even if it is not mandatory, I suggest aligning with the abbreviations used in the current international guidelines, i.e. using 'ART' and not 'cART', and 'PWHIV' and not 'PLHIV', in both the abstract and the main text.

metabolic dysregulation: unclear what the authors mean, please specify

metabolic parameters : It is unclear what the authors mean. Please specify.

In the Results section of the abstract, the percentage variation and p values are given in brackets for some variables, the variation is written in the text for others, and only the p value is given in brackets for the remaining variables (without a measure of the variation, either expressed as perentage or mean change). If possible, I suggest standardising the way the results are expressed.

Conclusions: These findings underscore the importance of expanding HIV care to include proactive cardiometabolic surveillance and early intervention strategies aimed at preventing long-term vascular complications in PLHIV on cART.

The conclusions are not fully supported by the results, as the observed changes have not been interpreted. It is not possible to speculate whether proactive surveillance would be cost-effective, or whether the statistically significant changes observed could have no clinical impact within this 36-month period, to the extent that proactive surveillance could be justified for clinical purposes (as well as for the purposes of the study).

MAIN TEXT

Introduction:

“antiretroviral therapy, while essential for viral suppression and immune restoration, can induce adverse metabolic effects such as dyslipidemia, insulin resistance, and alterations in fat distribution, commonly referred to as HIV-associated lipodystrophy [3-5] “ These side effects do not apply to the antiretrovirals currently in use. In particular, none of the drugs counselled for first-line treatment can be blamed for causing lipodystrophy. Only PIs and efavirenz (which is not currently indicated for first-line treatment) can cause dyslipidaemia. The possibility of insulin resistance should also be explained more clearly, as this is not a general effect of all ARVs and is still being debated for many modern regimens, given that studies have produced conflicting results.

“ Among the key manifestations of lipodystrophy is visceral fat accumulation, which has been increasingly recognized as a driver of subclinical atherosclerosis in HIV-infected individuals. “ This is not applicable to modern ART. I suggest rewriting the introductory paragraph to refer to possible visceral fat accumulation outside the context of ART-induced lipodystrophy.

“[…] others emphasize the additive effect of HIV-specific factors, such as chronic inflammation and prolonged ART exposure.” This sentence needs references.

The aim of the study should be described at the end of the introduction. In the text: “ This study investigates the prevalence and potential risk factors associated with increased visceral fat and carotid intima- media thickness in HIV-infected patients receiving cART, aiming to further elucidate the complex relationship between HIV, treatment, and cardiometabolic health.”

The aim of the study should be rewritten. It is unclear whether it is intended to describe the prevalence of risk factors, in which case the introduction should focus on the specific risk factors to be investigated, or whether it is intended to “elucidate the complex relationship between HIV, treatment and cardiometabolic health”. If the latter, the aim is unclear and too generic and should be better explained. According to the abstract, the study appears to investigate longitudinal changes in fat accumulation and metabolic parameters 36 months after initiating first-line antiretroviral therapy (ART), while providing no data on the prevalence of any risk factors or the relationship between individual factors and HIV/ART.

I strongly suggest to avoid terms such as “HIV-infected”, as can be perceived as stigmatizing by readers and PWHIV.

In general, the introduction focuses on older ART and cardiovascular risk. It should instead focus on what is already known about metabolic changes at the beginning of ART , on what information is still lacking and which data gaps the study aims to fill.

Methods:

In study design and setting, it is stated that the study investigates “ metabolic changes “. While this terminology is acceptable in the introduction, in the methods section it would be better to avoid generic terms and list the precise variables to be investigated for the purposes of the study, distinguishing between primary and secondary objectives.

Study participants: according to the exclusion criteria, the estimated time of infection was not considered. If I have understood correctly, both advanced and recent/acute diagnoses could be included and analysed together. If this is true, this difference should be taken into account in the analysis, with people with low CD4 being considered separately, at least in a sensitivity analysis.

Also, there are no inclusion criteria based on the type of ART initiated as the initial treatment. Please specify how this possible confounding factor was taken into account and which first-line treatments are used in your centre in accordance with the study protocol/local guidelines.

Finally, virological suppression should be evaluated at the study timepoints to determine whether people are actually taking ART.

Line 105, 146, and elsewhere in the text. If possible, avoid the term 'patients' and use 'study participants', 'participants' or 'people' instead, in line with 'people-first' language.

Line 106: “sociodemographic characteristics, HIV disease history, other comorbid conditions, health-related behaviors, current medication use for hypertension, diabetes, and dyslipidemia” this information is never given in results.

Line 137: “Changes in selected variables from baseline to the first follow-up were assessed using the Wilcoxon signed-rank test and McNemar's test.” Is this referred to as the 12-month follow-up? Which variables were studied using this method and which ones using the mixed model? Why did you choose different statistical evaluations for different variables? Please explain.

Also: How did you select the variables to adjust the final model?

Results:

Line 147, 156, and elsewhere in the text: As mentioned before, I strongly suggest avoiding the use of terms such as 'HIV-infected' (see previous comments on people-first language)

line 150: “mean nadir viral load” : This is probably the peak (rather than the nadir) of the viral load. Please verify whether the distribution of the viral load is normal, or whether it would be more informative to provide the median value.

Line 160-161 “ These findings indicate a progressive accumulation of central adiposity associated with long-term cART exposure. (Table 1)” This conclusion is not fully supported by the data. If there is indeed a statistically significant change in BMI over the three-year observation period, this could be associated with treatment initiation or even with ageing over three years. I cannot see the amount of weight gain in the table, so it is difficult to determine whether this is comparable with the expected weight gain in a similar age group in the general population or whether it is indeed higher, as expected. In both cases, weight gain is expected in ART-naïve individuals initiating first-line treatment, and this factor (rather than 'long-term' ART exposure) is probably implicated, as the results of a three-year observation period would probably be different in a population treated for 10 years. In conclusion, the data cannot support the statement, which should either be reworded or moved to the discussion section, where it is appropriate to speculate on the results.

Table 1 requires a legend for the abbreviations used, as well as an indication of whether the values shown for each variable are the mean or the median.

HOMA-beta: the change does not appear to be statistically significant, according to the p-values given in Table 1 (only borderline significance at 24 months; not observed at months 12 or 36). However, the authors speculate in the Results section that this parameter is indicative of 'impaired insulin secretion and early disruption of glucose homeostasis'. This speculation should be included in the discussion rather than the results. Moreover, I do not understand whether the hypothesis is based solely on the borderline significance of the change in this index at 24 months (p = 0.049), or whether this is rather justified by the clinically significant cut-off point for these values derived from the literature review. Please specify this (in the discussion, not in the results.)

Line 166 “reaching levels consistent with elevated adiposity” . The term 'elevated adiposity' is qualitative and not informative. If you used cut-off values in your study to define elevated adiposity, please specify them in the Methods section with adequate referencing. Otherwise, avoid qualitative judgements in the results, where the data should be reported objectively, without opinion or comment. These should be reserved for the discussion section.

Line 167-168: A definition of android and gynoid fat should be added to the methods section. Also, the implications of an increase in the android-to-gynoid fat ratio are not adequately stated in the introduction or methods.

Line 176: “Fasting glucose levels increased significantly over time (p < 0.05).“ By how much did the level increase? The data is not shown in either the text or the table. Please add it to enable readers to evaluate the clinical significance of the increase (beyond statistical significance).

Line 184 “ Renal function, assessed by estimated glomerular filtration rate (eGFR), showed a significant decline over time (p < 0.05) after cART initiation. However, albuminuria did not change significantly. “ The eGFR calculation, which is based on creatinine levels, can be affected by renal transporters and their interactions with many modern antiretrovirals. Although this results in an apparent GFR reduction, there is usually no real impairment. This evaluation is closely linked to the type of antiretrovirals used in this study population, which cannot be omitted from such a study. In any case, the interpretation of the results should be moved to the discussion section.

Line 202: “ These findings suggest a gradual and widespread progression of subclinical atherosclerosis during long-term cART.“ I suggest moving this interpretation to the discussion section.

Table 2: The use of abbreviations is redundant as full definitions are provided. You can either remove the abbreviations or, if you prefer, leave them in the table and move their definitions to the table legend.

In the text and Table 3, please specify whether correlations have been tested for baseline values or variations at any study timepoints.

Discussion

In the first sentence of the discussion “ This prospective cohort study provides important insights into the longitudinal effects of combination antiretroviral therapy (cART) on metabolic health”, I suggest replacing 'metabolic health' with 'metabolic parameters', as the changes described in the results do not directly equate to better or worse 'health' compared to baseline values.

In fact, metabolic health is never studied specifically: the authors never establish a threshold for any of the variables studied that could be linked to a pathological interpretation. For example, can the variation of mean BMI from 23.0 to 23.7 be considered an indicator of poor metabolic health? Or could it be the change in fasting glucose without any change in HbA1c or HOMA index? I understand that the authors' interpretation is that an increase in adipose tissue could be problematic, but there are actually no pathological cut-offs that have been exceeded by the study participants, including those evaluated for IMT. If, instead, predefined cut-offs based on the literature were exceeded by the values recorded in the study participants, please specify them in the methods (with references) and discuss them in the discussion.

Line 232: “This redistribution of fat is clinically relevant due to its strong association with metabolic dysfunction “ This sentence needs referencing. The context should also be better explained. According to literature data, which level of fat accumulation should be considered detrimental, and how applicable are these data to the study population?

Line 233 and following: “Our findings confirm this link, showing significant increases in fasting glucose levels and a decline in HOMA-Beta, suggesting reduced pancreatic beta-cell function and early disturbances in glucose homeostasis. While traditional markers such as HbA1c, lipid profiles, and HOMA-IR remained relatively stable, the subtle metabolic shifts observed emphasize the importance of early monitoring and intervention to mitigate long-term risk. “

If the authors feel that fasting glucose levels are important, they should present their baseline and follow-up values in the results. The authors might also reconsider the role of HOMA beta, as this value differs from baseline only at 24 months and not at 36 months. Thus, their hypothesis that early changes could predict long-term risk is not supported by their own results. This is also suggested by the fact that the other parameters considered (namely HbA1c and the HOMA index) do not change. Why investigate all these factors if, in the end, only fasting glucose (whose values are not shown) is used to draw conclusions?

Line 238: “despite the changes in adiposity and glucose metabolism, the prevalence of metabolic syndrome did not significantly increase, possibly due to the relatively young age and preserved immune status of the cohort at baseline. “ This sentence supports my previous objection that, according to your data, glucose metabolism did not change significantly. Also, the increase in fat should be contextualised more clearly, as you showed in the results that underweight people were also included in the analysis, and the 'return to health' phenomenon could explain your results at least in part. Weight and fat increases are not always negative, and the lack of correlation with an increase in metabolic syndrome could support this, even if the follow-up period is too short to be sure that there will be no metabolic consequences in the long term.

Line 240: “the observed decline in estimated glomerular filtration rate (eGFR) suggests that renal function may be vulnerable to the long-term effects of cART, even in the absence of overt clinical markers such as albuminuria.” This sentence needs referencing.

Also, many antiretrovirals can cause an apparent increase in creatinine levels without affecting the actual glomerular filtration level (e.g. integrase inhibitors, cobicistat-boosted drugs and rilpivirine). ART regimens should be mentioned somewhere in the text to enable the results to be evaluated more effectively. It would also be useful to know how many people were treated with potentially nephrotoxic drugs such as TDF. Otherwise, the evaluation of eGFR can be omitted from the results, as it is neither informative nor interpretable in its current form.

Line 243 and following: What is the IMT cut-off that correlates with unfavourable study outcomes? Have these values been reached by the study populations? Is there data on unfavourable outcomes with values similar to those recorded in this study?

Study limitations: This paragraph should cover the limitations that I mentioned in my previous comments.

Conclusions

The authors conclude that they observed “fat accumulation, metabolic dysregulation, and progression of carotid atherosclerosis”. While I agree that they observed fat accumulation and progression of IMT, I disagree that they observed metabolic dysregulation (cholesterol, LDL, triglycerides and insulin resistance did not change) or clinically significant atherosclerosis (only subclinical, if I have correctly understood the results).

Reviewer #2: The study addresses an important clinical issue by investigating the progression of carotid intima-media thickness (IMT), visceral fat accumulation, and metabolic alterations in HIV-positive patients within a prospective cohort. However, several methodological and reporting weaknesses significantly undermine the strength of the findings and should be carefully addressed.

The description of the IMT measurement protocol in the Methods section is unclear. It is not specified whether IMT was assessed exclusively in the common carotid artery (CCA), specifically 3 cm below the carotid bulb, or also at the bifurcation and internal carotid artery (ICA). Although Table 2 clearly reports the different carotid segments where IMT was measured, this information should also be explicitly described in the Methods to ensure clarity and reproducibility.

Moreover, the manuscript does not indicate whether the IMT measurements were performed by a single operator or multiple operators, nor whether they were blinded to previous measurements or clinical characteristics. This omission raises concerns about potential measurement bias and inter-operator variability.

A particularly important limitation is the absence of automated IMT measurement. Considering that some of the reported differences are below 0.1 mm, it is highly questionable whether such minimal changes can be reliably detected through manual evaluation. This should be clearly acknowledged and discussed as a major limitation, given its impact on the validity of the results.

In Table 2, many p-values are reported as exactly 1.000, which is statistically unusual. It is unclear whether this reflects a rounding convention, a specific statistical method, or a possible issue in data processing. Clarification is needed to understand the robustness of the statistical analyses.

Another relevant point concerns the use of preventive therapies. The manuscript does not specify whether patients were allowed to initiate anti-atherosclerotic treatments (e.g., statins, anti-hypertensive agents) during the follow-up period, nor whether any such changes were tracked or adjusted for in the analysis. This represents a potentially significant confounding factor that must be addressed.

Finally, it is not explained which IMT value was used for the univariate correlation analyses — whether a single-site value, an average, the maximum IMT, or a composite index. This information is essential to properly interpret the results and ensure reproducibility.

Overall, the study explores a highly relevant topic, but the current version of the manuscript presents several methodological limitations that need to be clarified and discussed in detail before the findings can be considered robust.

**Do you want your identity to be public for this peer review?** For information about this choice, including consent withdrawal, please see our Privacy Policy

Reviewer #1: No

Reviewer #2: No

---

## [Author Response · Author response to Decision Letter 1]

9 Sep 2025

Manuscript title: Progression of Carotid Intima-Media Thickness, Visceral Fat Accumulation, and Metabolic Derangement in People Living with HIV Initiating Antiretroviral Therapy: A Prospective Cohort Study at Thailand’s Tertiary Care Center

Journal: PLOS ONE

Manuscript ID: [PONE-D-25-27072] - [EMID:c21e87286d82c702]

We sincerely thank the Academic Editor and both Reviewers for their thoughtful and constructive feedback on our manuscript. We have carefully revised the paper in response to all comments. Below we provide a detailed point-by-point response, with reviewer comments in italics and our replies directly beneath. All revisions have been incorporated into the updated manuscript, and we have uploaded a tracked-changes version as requested.

⸻

Reviewer #1

1. Abstract – Use of abbreviations (“cART”, “PLHIV”).

Comment: Align with international guidelines, use “ART” and “PWHIV.”

Response: Revised throughout abstract and main text, replacing “cART” with “ART” and “PLHIV” with “PWHIV.”

Revision: Abstract p.1; throughout manuscript.

2. Abstract – Clarity of “metabolic dysregulation / metabolic parameters.”

Comment: Please specify what is meant.

Response: Clarified to specify fasting glucose, HOMA-β, lipid profile, and renal function. Replaced “metabolic dysregulation” with “early metabolic alterations.”

Revision: Abstract Results/Conclusions; Introduction p.2.

3. Abstract – Inconsistent reporting of % change and p-values.

Comment: Standardize.

Response: Standardized to present mean change with % change and p-values consistently.

Revision: Abstract Results; Results; Tables 3–4.

4. Abstract – Conclusions overstated.

Comment: Proactive surveillance not supported.

Response: Rephrased conclusion: “These findings underscore the need for monitoring cardiometabolic risk in PWHIV on ART.”

Revision: Abstract Conclusions p.1.

5. Introduction – Lipodystrophy and outdated ARVs.

Response: Revised to distinguish historical lipodystrophy from modern ART; clarified visceral fat accumulation remains a concern. Added references.

Revision: Introduction p.2.

6. Aim of study unclear.

Response: Revised aim: “This study aimed to characterize 36-month longitudinal changes in fat distribution, metabolic parameters, and carotid intima-media thickness (cIMT) following ART initiation in Thai adults.”

Revision: Introduction, final paragraph.

7. Language – people-first.

Response: Revised throughout to “people living with HIV.”

8. Methods – “metabolic changes” vague.

Response: Specified primary outcomes (fat distribution, cIMT) and secondary outcomes (glycemia, lipids, renal indices).

Revision: Methods p.3.

9. Methods – Heterogeneous population.

Response: Acknowledged as limitation; stratification not performed due to modest sample size.

Revision: Limitations p.27.

10. Methods – ART regimens not described.

Response: Added Table 1 summarizing ART class exposure; discussed class-specific effects.

Revision: Table 1; Discussion p.25–26.

11. Methods – Virological suppression not shown.

Response: Added viral load suppression and CD4 recovery (Table 2).

Revision: Results Table 2.

12. Methods – “patients” vs “participants.”

Response: Revised consistently to “participants.”

13. Methods – Sociodemographic characteristics missing.

Response: Not systematically collected; acknowledged as limitation.

Revision: Limitations p.27–28.

14. Methods – Statistical approach unclear.

Response: Expanded: Wilcoxon/McNemar for baseline→12m; GLMM for longitudinal trends; covariates chosen a priori.

Revision: Methods p.8–9.

15. Results – Weight gain attribution.

Response: Revised to report neutrally; discussion of causes moved to Discussion.

Revision: Discussion p.24–25.

16. Results – HOMA-β significance.

Response: Rewritten to note transient decline at 24m only; cautious interpretation.

Revision: Discussion p.25.

17. Results – “Elevated adiposity.”

Response: Removed qualitative phrasing; reported numeric data only.

Revision: Results p.13.

18. Results – Fasting glucose values not shown.

Response: Added absolute values in Table 3.

19. Results – Renal function interpretation.

Response: Interpretation moved to Discussion; noted possible ART–transporter effects.

Revision: Discussion p.25–26.

20. Results – Clarify correlation values.

Response: Clarified correlations based on longitudinal change values.

Revision: Results/Table 5 legend.

21. Discussion – “Metabolic health” wording.

Response: Revised to “metabolic parameters.”

22. Discussion – Clinical cut-offs missing.

Response: Added cut-offs (A/G >1.0; cIMT >0.8 mm) and references; clarified most remained below thresholds.

Revision: Methods; Discussion p.8.

23. Discussion – HOMA-β overstated.

Response: Rewritten to reflect transient, not persistent, dysfunction.

24. Discussion – Weight gain may reflect return-to-health.

Response: Added discussion of return-to-health phenomenon.

25. Limitations – Expanded.

Response: Added regimen heterogeneity, lack of control group, transporter effects, manual IMT measurement, unmeasured confounders.

26. Conclusions – Overstated.

Response: Revised to: “Long-term ART was associated with fat accumulation and progressive cIMT thickening, with modest early metabolic changes.”

⸻

Reviewer #2

1. IMT measurement description incomplete.

Response: Revised Methods to specify CCA, bifurcation, ICA sites.

2. Operator blinding/variability not described.

Response: Added that scans were performed by blinded radiologists; reproducibility not formally assessed (limitation).

3. Manual vs automated IMT.

Response: Acknowledged as limitation.

4. P-values reported as 1.000.

Response: Clarified rounding method; noted in Methods.

5. Preventive therapies not reported.

Response: Not systematically recorded; acknowledged as limitation.

6. IMT correlation analyses – clarify value.

Response: Clarified use of right carotid bifurcation mean values.

⸻

Editorial Requirements Addressed

• Financial disclosure: Revised to include funder statement.

• Ethics statement: Expanded with IRB full name, approval code, and written informed consent.

• Formatting: Revised to follow PLOS ONE templates and LaTeX requirements.

• Terminology: Standardized to ART, PWHIV, and “participants.”

All reviewer and editorial comments have been fully addressed, and the manuscript has been revised accordingly.

---

## [Decision Letter · Decision Letter 1]

8 Oct 2025

Progression of Carotid Intima-Media Thickness, Visceral Fat Accumulation, and Metabolic Derangement in People Living with HIV Initiating Antiretroviral Therapy: A Prospective Cohort Study at Thailand’s Tertiary Care Center

PONE-D-25-27072R1

Dear Dr. Boonsaen,

We’re pleased to inform you that your manuscript has been judged scientifically suitable for publication and will be formally accepted for publication once it meets all outstanding technical requirements.

Kind regards,

Giuseppe Vittorio De Socio, MD, PhD

Academic Editor

PLOS ONE

Additional Editor Comments (optional):

Reviewers' comments:

Reviewer's Responses to Questions

**Comments to the Author**

Reviewer #1: All comments have been addressed

Reviewer #2: All comments have been addressed

2. Is the manuscript technically sound, and do the data support the conclusions?

Reviewer #1: Yes

Reviewer #2: Yes

3. Has the statistical analysis been performed appropriately and rigorously?

Reviewer #1: I Don't Know

Reviewer #2: Yes

4. Have the authors made all data underlying the findings in their manuscript fully available?

Reviewer #1: No

Reviewer #2: Yes

5. Is the manuscript presented in an intelligible fashion and written in standard English?

Reviewer #1: Yes

Reviewer #2: Yes

Reviewer #1: (No Response)

Reviewer #2: The manuscript has been improved according to the reviewers’ suggestions. I have no further comments.

**Do you want your identity to be public for this peer review?** For information about this choice, including consent withdrawal, please see our Privacy Policy

Reviewer #1: No

Reviewer #2: No

---

## [Editor Report · Acceptance letter]

PONE-D-25-27072R1

PLOS ONE

Dear Dr. Boonsaen,

I'm pleased to inform you that your manuscript has been deemed suitable for publication in PLOS ONE. Congratulations! Your manuscript is now being handed over to our production team.

Kind regards,

on behalf of

prof. Giuseppe Vittorio De Socio

Academic Editor

PLOS ONE